# Comprehensive analysis of nasal IgA antibodies induced by intranasal administration of the SARS-CoV-2 spike protein

Kentarou Waki[1], Hideki Tani[2], Eigo Kawahara[3,4], Yumiko Saga[2], Takahisa Shimada[2], Emiko Yamazaki[2], Seiichi Koike[5], Yoshitomo Morinaga[3,4], Masaharu Isobe[4,5], Nobuyuki Kurosawa[4,5]*

[1]Laboratory of Molecular and Cellular Biology, Graduate School of Science and Engineering for Education, University of Toyama, Toyama, Japan; [2]Department of Virology, Toyama Institute of Health, Toyama, Japan; [3]Department of Microbiology, Toyama University Graduate School of Medicine and Pharmaceutical Sciences, Toyama, Japan; [4]Center for Advanced Antibody Drug Development, University of Toyama, Toyama, Japan; [5]Laboratory of Molecular and Cellular Biology, Graduate School of Innovative Life Science, University of Toyama, Toyama, Japan

## eLife Assessment

This work provides **important** insights into mucosal antibody responses against SARS-CoV-2 following intranasal immunization by characterizing a large number of monoclonal antibodies at both mucosal and non-mucosal sites. The evidence supporting the claims is **solid**. The demonstrated in vitro antiviral activity of antibodies characterized provides a rationale for developing mucosal vaccines, especially if confirmed in vivo and benchmarked against antibodies generated following intramuscular vaccination.

*For correspondence:
kurosawa@eng.u-toyama.ac.jp

**Competing interest:** The authors declare that no competing interests exist.

**Abstract** Intranasal vaccination is an attractive strategy for preventing COVID-19 disease as it stimulates the production of multimeric secretory immunoglobulin A (IgA), the predominant antibody isotype in the mucosal immune system, at the target site of severe acute respiratory syndrome coronavirus 2 (SARS-CoV-2) entry. Currently, intranasal vaccine efficacy is evaluated based on the measurement of polyclonal antibody titers in nasal lavage fluid. However, how individual multimeric secretory IgA protects the mucosa from SARS-CoV-2 infection remains to be elucidated. To understand the precise contribution and molecular nature of multimeric secretory IgA induced by intranasal vaccines, we developed 99 monoclonal IgA clones from nasal mucosa and 114 monoclonal IgA or IgG clones from nonmucosal tissues of mice that were intranasally immunized with the SARS-CoV-2 spike protein. The nonmucosal IgA clones exhibited shared origins and common and unique somatic mutations with the related nasal IgA clones, indicating that the antigen-specific plasma cells in the nonmucosal tissues originated from B cells stimulated at the nasal mucosa. Comparing the spike protein binding reactivity, angiotensin-converting enzyme-2-blocking, and in vitro SARS-CoV-2 virus neutralization of monomeric and multimeric secretory IgA pairs recognizing different epitopes showed that even non-neutralizing monomeric IgAs, which represent 70% of the nasal IgA repertoire, can protect against SARS-CoV-2 infection when expressed as multimeric secretory IgAs. We also demonstrated that the intranasal administration of multimeric secretory IgA delivered as prophylaxis in the hamster model reduced infection-induced weight loss. Our investigation is the

first to demonstrate the function of nasal IgA at the monoclonal level, showing that nasal immunization can provide effective immunity against SARS-CoV-2 by inducing multimeric secretory IgAs at the target site of the virus infection.

## Introduction

Immunoglobulin A (IgA) is differentially distributed between the systemic and mucosal immune systems (*Li et al., 2020*). Monomeric IgA (M-IgA) is predominantly present in serum, whereas secretory IgA (S-IgA) is the most prevalent in mucous secretions. S-IgA comprises two IgA units and one J-chain, which are synthesized and assembled in local plasma cells. Secretory components expressed on the basolateral surface of mucosal epithelial cells bind the IgA complex through the J-chain and transport the molecule to the apical cell membrane (*Woof and Kerr, 2006*). Recent studies have shown that S-IgA in the nasal mucosa exists not only as dimers but also as trimers and tetramers in the human upper respiratory mucosa (*Saito et al., 2019*). S-IgA plays an important role in the protection and homeostatic regulation of the airway, intestinal and vaginal epithelium through a process known as immune exclusion and immunosuppression (*Matsumoto, 2022*). Recent studies have demonstrated that multimeric S-IgA is more effective and provides greater cross-protection than IgG and M-IgA (*Okuya et al., 2020b*; *Asahi et al., 2002*; *Dhakal et al., 2018*; *Asahi-Ozaki et al., 2004*; *Wang et al., 2021*).

COVID-19 is a disease caused by infection with severe acute respiratory syndrome coronavirus 2 (SARS-CoV-2) that causes mild upper respiratory symptoms in most cases, but some patients develop bilateral pneumonia with acute respiratory distress (*Budinger et al., 2021*). The SARS-CoV-2 spike protein expressed on the virus surface is a multidomain homotrimer protein composed of an S1 domain consisting of an N-terminal domain (NTD) and receptor binding domain (RBD) and an S2 domain that mediates fusion of the virus and host cell membrane. Viral infection is initiated through the interaction between the RBD and the host receptor angiotensin-converting enzyme-2 (ACE2) (*Jackson et al., 2022*). Thus, the spike protein is the main target for current vaccine development because antibodies against this protein can neutralize the infection. Currently approved intramuscular COVID-19 vaccines predominantly induce a systemic immune response by producing IgG in serum before they cause severe tissue damage, resulting in a high degree of efficacy (*Polack et al., 2020*). However, these vaccines mainly induce IgG and M-IgA in serum; they do not induce S-IgA, which coats the upper respiratory tract mucosal surface (*DeFrancesco, 2020*). The emergence of highly infectious SARS-CoV-2 omicron variants may undermine the therapeutic efficacy of vaccines, requiring more effective vaccination to prevent SARS-CoV-2 infection. Since nose epithelial cells are a primary target for SARS-CoV-2, intranasal vaccinations that induce S-IgA in the upper respiratory tract are desirable for protection against the infection and transmission of the virus (*Sungnak et al., 2020*). To date, some intranasal vaccines are under development and have shown robust mucosal and humoral immune responses in animal models (*Houston, 2023*; *Chavda et al., 2021*; *Alu et al., 2022*; *Barrett et al., 2021*; *Bricker et al., 2021*; *D'Arco et al., 2021*; *Kim et al., 2021*; *Vabret et al., 2020*; *Ohtsuka et al., 2021*). However, there have been few published clinical trials of intranasal SARS-CoV-2 vaccines on humans (*Ewer et al., 2021*; *Zhu et al., 2023*). To evaluate the response to intranasal vaccines, researchers have utilized nasal lavage fluids containing polyclonal S-IgA (*Maltseva et al., 2022*; *Gianchecchi et al., 2019*; *Wong et al., 2022*; *Afkhami et al., 2022*; *Sui et al., 2021*; *Azzi et al., 2022*). However, obtaining information about the functions of individual antibodies in the polyclonal antiviral mucosal repertoire is challenging due to the broad nature of such assays. It has also been shown that intranasal vaccination induces a systemic antibody response, but the origin of these antibodies has not previously been investigated at the molecular level (*Tumpey et al., 2001*; *Zhu et al., 2023*). Thus, intranasal vaccines require novel approaches to evaluate the quality and quantity of IgA response.

To understand the precise contribution and molecular nature of S-IgA induced by intranasal vaccines in relation to its antiviral function, it is essential to develop monoclonal S-IgAs from plasma cells localized in the nasal mucosa and study their function. However, because of the difficulties in developing monoclonal IgA antibodies from nasal mucosal tissue, many studies have used artificial switching of IgG to IgA for recombinant production, and this approach has been used to study the protective properties of IgA against pathogens (*Ejemel et al., 2020*; *Saito et al., 2019*). Such technological

limitations have hampered the biochemical and clinical evaluation of intranasal vaccination at the molecular level.

Our group has established robust protocols for isolating single antigen-specific plasma cells from a variety of animals and the high-throughput production of monoclonal antibodies (*Kurosawa et al., 2012*). Here, using this technology, we developed 213 antigen-specific monoclonal antibodies from plasma cells derived from mice that were intranasally immunized with a stabilized SARS-CoV-2 spike trimer protein (Spike) and demonstrated that intranasal immunization induced potentially nasal-derived antibodies in the spleen, lung, and blood. Analysis of the properties of monoclonal IgAs recognizing different epitopes revealed that multimerization of M-IgAs could induce neutralizing activity in vitro and in vivo. These results are the first of their kind to be demonstrated by obtaining monoclonal S-IgAs from nasal mucosal tissue. Our data provide insights into the nasal immunity that may help shape future intranasal vaccine development.

## Results
### Development of anti-spike monoclonal M-IgAs from plasma cells localized in nasal mucosa

To evaluate the immune response after the intranasal administration of the SARS-CoV-2 spike protein of the Wuhan-Hu-1/D614G strain (Spike$_{Wuhan}$), nasal lavage fluids and serum were collected from the mice 1 week after the last immunization. The intranasal administration of Spike$_{Wuhan}$ induced high levels of antigen-specific IgA but not IgG in nasal lavage fluids compared with those of the phosphate-buffered saline (PBS) control (*Figure 1A*).

Elevated levels of antigen-specific IgA and IgG responses were detected in serum from the immunized mice, consistent with a prior study demonstrating that intranasal vaccinations induce both nasal and serum IgA levels (*Sterlin et al., 2021*; *Maltseva et al., 2022*). We selected three immunized mice with high nasal IgA titers and isolated antigen-specific plasma cells from the nasal mucosa, spleen, lung, and blood. Cells isolated from these tissues were stained with fluorescently labeled anti-CD138, ER-Tracker Blue-White DPX (ER-Tracker), anti-IgA, and S1 domain of Spike$_{Wuhan}$ (S1), and then antigen-specific plasma cells defined as anti-CD138$^+$, ER-Tracker$^{high}$, anti-IgA$^+$, and S1$^+$ were isolated by fluorescence-activated cell sorting (FACS). ER-Tracker is a fluorescent dye that is highly selective for the endoplasmic reticulum of living cells. Because plasma cells have an expanded endoplasmic reticulum for properly folding and secreting large quantities of antibodies, using ER-Tracker along with anti-CD138 facilitates the isolation of plasma cells from lymphocytes without the need for additional antibodies (*Kurosawa et al., 2012*). Consistent with the pattern of the IgA response observed in nasal lavage fluid, the proportions of antigen-specific IgA$^+$ plasma cells in the nasal mucosa of immunized mice were significantly higher than those of control mice (*Figure 1B*, *Figure 1—figure supplement 1*).

Cognate pairs of immunoglobulin heavy chain variable (VH) and kappa light chain variable (VL) genes were amplified by rapid amplification of 5' cDNA ends PCR from the single-sorted cells. After constructing full-length immunoglobulin heavy and kappa light chain genes, antibodies were expressed by DNA transfection into CHO-S cells. Antigen-specific antibody clones were identified by enzyme-linked immunosorbent assay (ELISA) with immobilized Spike$_{Wuhan}$. After sequencing the entire coding region of heavy and light chain genes, the antibodies sharing the same V-(D)-J genes were grouped (*Supplementary file 1*). Then, representative antibody clones from each group were analyzed for their binding to the spike protein RBD of Wuhan, Beta, Kappa, and Delta variants and an NTD of Spike$_{Wuhan}$. The antibodies were also analyzed for their ability to block ACE2 binding to RBDs and to neutralize the Wuhan pseudotyped virus. Based on their properties, each group of antibodies was arbitrarily categorized into five types. Type 1: anti-RBD, ACE2 blocking neutralizing antibody; Type 2: anti-RBD, *ACE2 blocking* non-neutralizing *antibody*; Type 3: anti-RBD, *non-ACE2 blocking* non-neutralizing *antibody*; Type 4: anti-NTD, non-ACE-2 blocking non-neutralizing *antibody*; and Type 5: *non-ACE-2 blocking* non-neutralizing antibody targeting epitopes other than RBD and NTD.

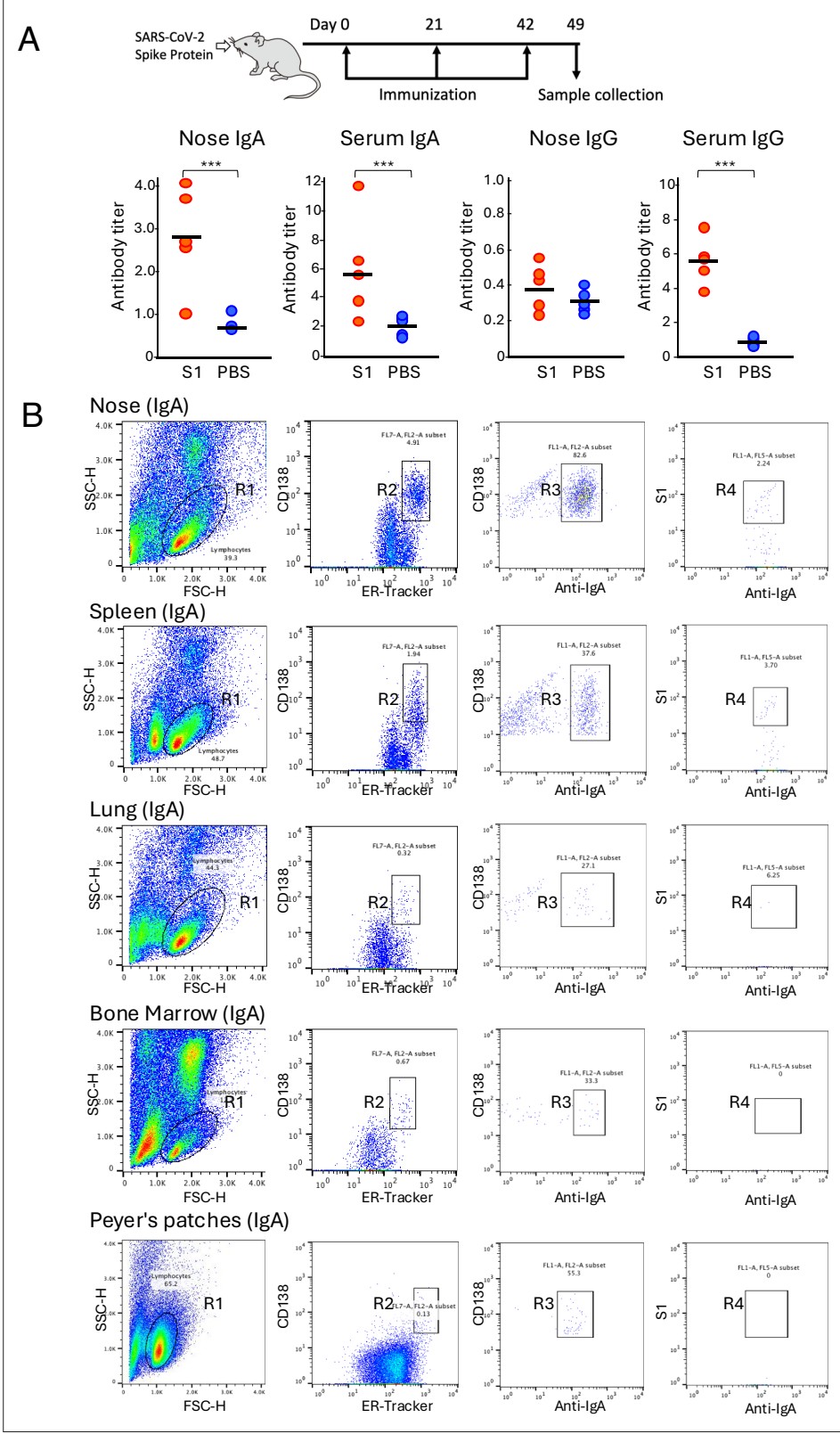

**Figure 1.** Isolation of antigen-specific IgA[+] plasma cells from mice that were intranasally immunized with Spike_{Wuhan}. (**A**) Mice were inoculated intranasally with 10 µg of Spike_{Wuhan} and 1 µg of cholera toxin as an adjuvant in 10 µl of phosphate-buffered saline (PBS), delivering the fluid dropwise into the nostril a total of three times at 3-week intervals. Nasal lavage fluid and serum were collected from the mice 1 week after the last immunization,

*Figure 1 continued on next page*

*Figure 1 continued*

and antibody responses were evaluated using enzyme-linked immunosorbent assay (ELISA) (n=5) (∗∗∗p<0.001) . The antibody titers are expressed as optical density ($OD_{450}$) value per total protein in nasal lavage fluids or serum. (**B**) Fluorescence-activated cell sorting (FACS) gating strategy for the isolation of S1-specific plasma cells from mice. Plots represent the sequential gating strategy. Lymphocytes (R1 gate) were stained with anti-CD138 and ER-Tracker to enrich plasma cells (CD138+ ER-Tracker^High fraction, R2 gate). IgA+ plasma cells gated in R3 were selected from the R2-gated plasma cell fraction by staining with anti-IgA antibody. The antigen-specific plasma cells gated in R4 were further selected from the R3-gated Ig A+ plasma cell fraction by staining with S1 domain of Wuhan severe acute respiratory syndrome coronavirus 2 (SARS-CoV-2) spike protein (**S1**). The numbers indicate the percentages of cells in the gated area. A total of 100,000 events were recorded. Representative data from the No. 1 mouse are shown.

The online version of this article includes the following figure supplement(s) for figure 1:

**Figure supplement 1.** Isolation of antigen-specific IgA+ plasma cells from mice that were intranasally immunized with Spike_Wuhan.

## Intranasal immunization induces functionally diverse IgA in the nasal mucosa and spleen

We conducted a detailed characterization of antibodies obtained from the No. 1 mouse, as many antigen-specific M-IgA clones were obtained. Of the 51 nasal M-IgA clones analyzed, they were classified into 11 groups based on their V-(D)-J usage, with the majority (83%) clonally expanding into four major clusters (G2, G3, G4, and G10) and the remainder (17%) scattered across branches (*Figure 2A and B*).

## Intranasal immunization induces potentially nasal-derived antibodies in the spleen, lung, and blood

Previous reports suggest that class-switched, affinity-matured B cells selected by NALT migrate to the regional lymph node and then return to the nasal mucosa and nonmucosal tissues to differentiate into plasma cells (*Shimoda et al., 2001*). If this is the case, a fraction of S1-specific plasma cells differentiated from nose-originated B cells may reside in the spleen and produce antibodies. To directly evaluate the cellular origin of the anti-S1 antibodies in nonmucosal tissues, we analyzed the presence of S1-specific plasma cells in the spleen, lung, bone marrow, and Peyer's patches. FACS analysis of splenocytes harvested from No. 1 mice showed the presence of S1-specific IgA+ plasma cells, but the splenocytes harvested from control mice did not exhibit S1-specific IgA+ plasma cells (*Figure 1B*, *Figure 1—figure supplement 1*). Single-cell-based immunoglobulin gene cloning from the splenocytes of No. 1 mouse resulted in the successful production of 57 S1-specific monoclonal IgAs. DNA sequence analysis of these clones revealed significant clonal overlap between the nose and spleen (*Figure 2A and B*). Clonal overlap was found in the G2-G3, G5, G6, and G10 clusters, in which G10 possessed the most expanded splenic clones, as in the case of nasal IgA. Further investigation was conducted to determine if plasma cells expressing S1-specific IgG+ antibody, which share V-(D)-J with nasal IgA, could be identified in the spleen. Although the number of antigen-specific IgG+ plasma cells was limited, eight IgG clones specific for S1 were isolated, among which five were potentially nasal-derived. When we attempted to isolate antigen-specific IgA+ plasma cells from the lung, nine S1-specific IgA clones were obtained, among which three were potentially nasal-derived. The gut mucosa and bone marrow possessed a large population of IgA+ plasma cells. However, we could not detect S1-specific IgA+ plasma cells in Peyer's patches and bone marrow (*Figure 1B*).

Next, we focused on the expanded antibody groups G2, G3, and G10 to analyze the patterns of somatic hypermutation (SHM). We found complex patterns of shared and unique SHMs in antibodies obtained from the nose, spleen, and lung. In G2, the VHs of the three splenic clones (S530A, S208G, and S619A) had the same sequence, except for framework 4 of an IgG clone (S208G) and had three shared SHMs with the nasal clones (N219A and N226A). In addition, the VKs of the three splenic clones and N226A had the same sequence. In Group 3, seven nasal clones (N109A, N112A, N135A, N139A, N244A, N245A, and N715A) and two splenic clones (S545 and S607) were 100% identical in the nucleotide sequence, with N135A and N245 differing from the other clones by only one nucleotide in their VKs (*Figure 2C*). Sequence analysis of the most expanded group, Group 10, also demonstrated clonal overlap in the nose, lung, and spleen. For example, the VHs of a nasal clone (N221),

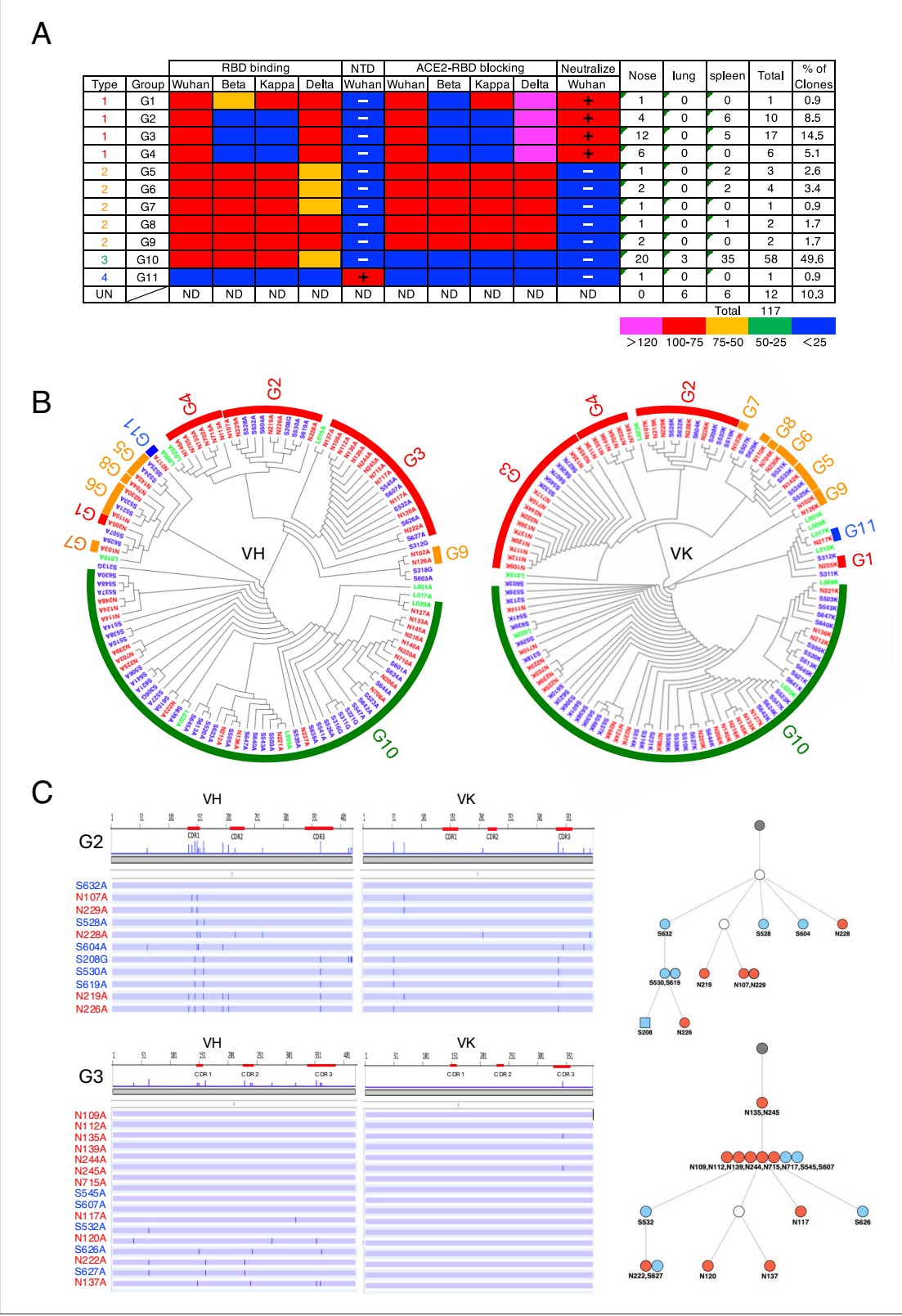

**Figure 2.** Intranasal immunization induces functionally diverse antibodies in the nasal mucosa and spleen. (**A**) Characterization of S1-specific monoclonal antibodies obtained from No. 1 mouse. The heatmap represents the relative intensity of antibody binding to receptor binding domains (RBDs) and blocking of the RBD-angiotensin-converting enzyme-2 (ACE2) interaction. Blue (0–25%), green (25–50%), orange (50–75%), and red (>75%). N-terminal domain (NTD) binding was considered positive (+) when the OD at 405 nm was >0.3 after the background was subtracted. Neutralizing

*Figure 2 continued on next page*

*Figure 2 continued*

activity was considered positive (+) when the antibody suppressed Wuhan pseudotyped virus infection by 50%. The figure reports values from a single experiment. UN, antibody type not determined. ND, antibody activity not determined. (**B**) Maximum-likelihood phylogenetic tree of the VH and VL chains of the S1-specific antibodies. Different colored fonts indicate antibodies obtained from the nose (red), spleen (blue), and lung (green). Bands on the outer ring indicate antibody groups. The color of the ring indicates antibody types: Type 1 (red), Type 2 (orange), Type 3 (green), Type 4 (blue), and Type 5 (gray). The antibody group is defined as clones using the same V-(D)-J usage and having an overall sequence identity of at least 95% from the signal peptide to framework 4 (FR4). The prefixes N, S, and L in the antibody clone numbers refer to antibodies derived from the nose, spleen, and lung, respectively. The suffixes A, G, and K in the antibody clone numbers refer to alpha, gamma, and kappa chains, respectively. (**C**) Nucleotide sequence alignment of VH and VL genes in the G2 and G3 antibodies from No. 1 mouse. The VH and VL sequences from the beginning of the signal peptide through the end of FR4 are shown as horizontal lines. Nucleotide changes relative to S632A and N109A are depicted as vertical bars across the horizontal lines. Different colored fonts indicate antibodies derived from the nose (red) and spleen (blue). Antibody phylogenetic trees based on VH/VK paired sequences are depicted. Gray circles represent the hypothetical germline configuration. White circles represent hypothetical ancestors. Colors indicate nasal (red) and splenic (blue) antibodies. Circles and squares indicate IgA and IgG, respectively.

The online version of this article includes the following figure supplement(s) for figure 2:

**Figure supplement 1.** Nucleotide sequence alignment of VH and VL genes in the G10 antibodies from No. 1 mouse.

spleen clones (S647, S503, S640, and S543), and a lung clone (L009) were 100% identical, with S647 differing from the other clones by only one nucleotide in VK (***Figure 2***, ***Figure 2—figure supplement 1***). Antibody lineage analysis of these clones suggests that each family of related antibodies originates from a common ancestor gene that was subjected to class switching and SHM. We also developed S1-specific antibodies from No. 2 and No. 3 mice and analyzed their sequences (***Supplementary file 1***).

In No. 2 mice, we obtained 29 nasal IgAs, 15 splenic antibodies (13 IgAs and 2 IgGs), 5 lung IgAs, and 6 blood IgAs. They were classified into 17 groups based on their V-(D)-J usage, and each group was categorized as either Type 1, 2, 3, 4, or 5 based on antibody properties (***Figure 3***). Potentially nasal-derived clones were found in the spleen, lung, and blood (G5, -6, and -8). These clones also showed a complex pattern of shared and unique SHM with nasal clones throughout the full length of the VH and VL genes. In mouse No. 3, we cloned 19 nasal IgAs and 22 splenic antibodies (17 IgAs and 5 IgGs). They were classified into 10 groups, and each group was categorized as either Type 1, 3, 4, or 5. A high degree of clonal overlap between the nose and spleen was found, in which 13 out of 22 splenic antibodies were potentially nasal-derived clones (***Figure 3***, ***Figure 3—figure supplement 1***). Analysis of the VH and VL repertoires of the three mice revealed that the expanded clones constituted varying fractions of the antibody repertoire among different mice despite having received the same antigen, and no group of antibodies stood out across mice, suggesting that individual mice had immunologically distinct responses (***Figure 3—figure supplement 2***). In all mice, only a few bound to the NTD, and most of them were RBD-directed. Approximately 30% of M-IgAs categorized as Type 1 showed neutralizing activity. Taken together, regarding the mutation frequency and the pattern of SHM, it can be assumed that B cells stimulated by nasal challenges were the major precursor of antigen-specific plasma cells in the spleen, lung, and blood, which may contribute to antibody production in the lower respiratory tract and systemic circulation.

## Multimerization of M-IgA enhances antigen-binding activity

To examine how IgA multimerization affects antigen-binding activity, four representative clones (N5203, N142, N114, and N217) were selected from each type of antibody, and S-IgAs were expressed by cotransfecting alpha heavy chain, kappa light chain, J-chain, and secretory component into CHO-S cells. Analysis of the purified S-IgAs by polyacrylamide gel electrophoresis (SDS-PAGE) revealed a band corresponding to the alpha heavy chain, kappa light chain, and secretory component. Native-PAGE analysis revealed that the antibodies comprised three quaternary structures, including a dimer (~400 kDa), a trimer (~550 kDa), and a tetramer (~750 kDa) at a molar ratio of 5:1:3 (***Figure 4A***).

Then, we compared the binding kinetics of each pair of M-IgA/S-IgA by surface plasmon resonance (SPR) with immobilized Spike$_{Wuhan}$, Spike$_{Delta}$, or Spike$_{Omicron}$. As shown in ***Figure 4***, N5203 M-IgA, which has a moderate binding affinity to Spike$_{Wuhan}$ (apparent equilibrium constant, $K_D$ = 5.2E-9), acquired dramatic enhancement of the binding activity to Spike$_{Wuhan}$ after multimerization ($K_D$ = 1.3E-13). However, multimerization did not enhance the binding activity to Spike$_{Delta}$ ($K_D$ = 2.1E-9). N142 M-IgA, which has a high affinity for Spike$_{Wuhan}$ ($K_D$ = 4.5E-11) and Spike$_{Delta}$ ($K_D$ = 7.8E-11), did not show

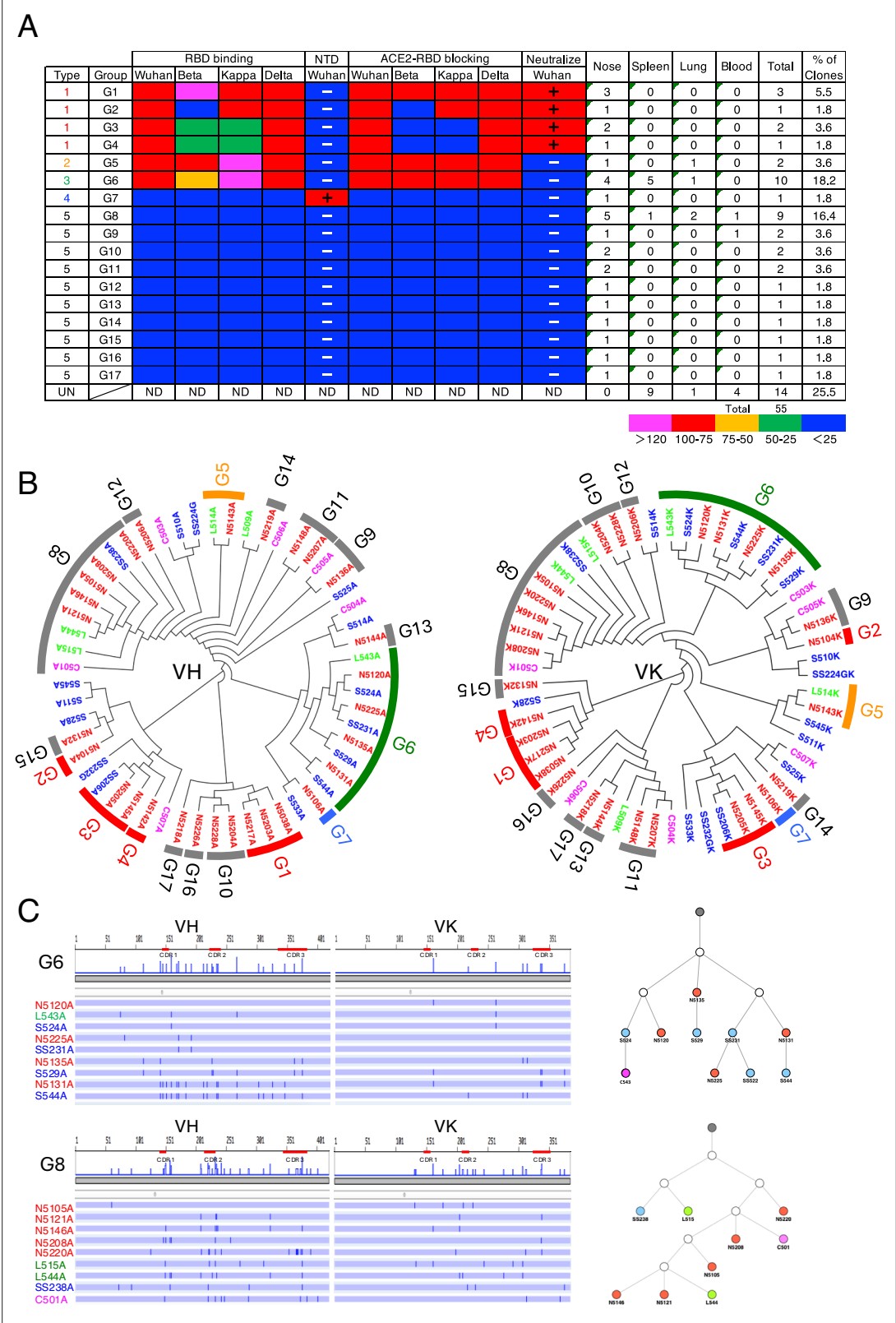

**Figure 3.** Characterization of S1-specific monoclonal antibodies obtained from No. 2 mouse. (**A**) A total of 51 S1-reactive antibodies were analyzed for their properties as in *Figure 2A*. UN, antibody type not determined. ND, antibody activity not determined. (**B**) Maximum-likelihood phylogenetic tree of the VH and VL chains of the S1-specific antibodies. Different colored fonts indicate antibodies obtained from the nose (red), spleen (blue), lung (green), and blood (magenta). Bands on the outer ring indicate antibody groups. The color of the band indicates antibody types: Type 1 (red), Type 2

*Figure 3 continued on next page*

*Figure 3 continued*

(orange), Type 3 (green), Type 4 (blue), and Type 5 (gray). The prefixes N, S, and L in the antibody clone numbers refer to antibodies derived from the nose, spleen, and lung, respectively. The suffixes A, G, and K in the antibody clone numbers refer to alpha, gamma, and kappa chains, respectively. (**C**) Nucleotide sequence alignment of VH and VL genes in the G6 and G8 antibodies from No. 2 mouse. The nucleotide changes relative to N5120A and N5105A are depicted as vertical bars across the horizontal lines. Different colored fonts indicate antibodies derived from the nose (red), spleen (blue), lung (green), and blood (magenta). Antibody phylogenetic trees based on VH/VK paired sequences are depicted. Gray circles represent the hypothetical germline configuration. White circles represent hypothetical ancestors. Colors indicate nasal (red), splenic (blue), lung (green), and blood (magenta) antibodies.

The online version of this article includes the following figure supplement(s) for figure 3:

**Figure supplement 1.** Characterization of S1-specific monoclonal antibodies obtained from No. 3 mouse.

**Figure supplement 2.** Analysis of the VH and VL repertoires of the three mice.

enhanced binding activity to them after multimerization. Almost the same phenomenon was found for N114 M-IgA, which has a high affinity for $Spike_{Wuhan}$ ($K_D$ = 6.8E-11) and $Spike_{Delta}$ ($K_D$ = 9.7E-11). N142 M-IgA, which has the lowest affinity to $Spike_{Wuhan}$ ($K_D$ = 2.1E-8), also acquired dramatic enhancement of the binding activity to $Spike_{Wuhan}$ after multimerization ($K_D$ = 6.4E-11) but not to $Spike_{Delta}$. All antibodies showed little or marginal levels of binding to $Spike_{Omicron}$. These results suggest that the degree of avidity of S-IgAs depends on the affinity of the parent monomeric antibody: antibodies with low or intermediate affinity in the monomeric state (N5203 and N217) can increase their avidity by multimerization but not antibodies with high affinity in the monomeric state (N142 and N114). These results are consistent with recent work by Saito et al., who examined the function of multimerized IgA against influenza viruses (***Saito et al., 2019***).

## Multimerization facilitates stronger neutralization activity in non-neutralizing M-IgA

Since ACE2-blocking activity is one indicator for evaluating the neutralizing activity of anti-SARS-CoV-2 antibodies, we next examined whether the multimerization of IgAs influences the RBD-ACE2 interaction by competitive ELISA (***Figure 5A***).

Although the multimerization of N5203 M-IgA led to the dramatic enhancement of $Spike_{Wuhan}$ binding, it only increased $RBD_{Wuhan}$-ACE2 blocking by 3.3-fold, and there were no differences in $RBD_{Delta}$-ACE2 blocking between M-IgA and S-IgA. In N142 M-IgA, multimerization slightly increased $RBD_{Wuhan}$-ACE2 blocking but not $RBD_{Delta}$-ACE2 blocking. Multimerization of N114 and N217 M-IgA did not enhance $RBD_{Wuhan}$-ACE2 and $RBD_{Delta}$-ACE2 blocking, as in the case of their monomeric forms. None of the antibodies showed $RBD_{Omicron}$-ACE2 blocking even at the highest antibody concentrations.

To analyze whether multimerization affects the functionality of M-IgAs, we tested all M-IgA/S-IgA pairs for their ability to neutralize pseudotyped lentiviruses bearing the Wuhan, Delta, or Omicron spike protein (***Figure 5B***). Both monomeric and multimeric forms of N5203 showed strong neutralizing activity against the Wuhan and Delta pseudotyped lentiviruses, but the concentrations required to achieve the 50% neutralizing titer ($NT_{50}$) were almost the same for both. Although both forms of N142 displayed $RBD_{Wuhan}$-ACE2 and $RBD_{Delta}$-ACE2 blocking, only the multimeric form showed neutralization activity against the Wuhan and Delta pseudotyped lentiviruses. Notably, N114 M-IgA and N217 M-IgA, which did not have $RBD_{Wuhan}$-ACE2 blocking activity, displayed neutralizing activity against the Wuhan but not the Delta pseudotyped lentivirus after multimerization. None of the antibodies exhibited neutralization activity against the Omicron pseudotyped lentivirus. We further tested the neutralization activity of these antibodies by using an authentic SARS-CoV-2 Wuhan strain (***Figure 5C***). N5203 S-IgA showed twofold higher neutralization activity than its monomeric counterpart. As predicted from the results of the pseudotyped lentivirus assay, multimerization of N142, N114, and N217 M-IgA induced neutralization activity, while their monomeric counterparts failed to neutralize live virus at the highest antibody concentration. These results suggest that the neutralizing activity of S-IgAs is not solely attributed to the spike protein affinity and the RBD-ACE2 blocking activity of M-IgAs, and that the valence and binding mode between the epitope and paratope affect their function.

Next, we selected three types of S-IgAs (N5203, N142, and N217) and investigated whether nasally administered S-IgAs could protect against Wuhan SARS-CoV-2 infection in a Syrian hamster model as a prophylactic treatment (***Figure 6***). The animals in the control group showed significant and progressive weight loss (20% by day 3) after the viral challenge. In contrast, animals treated with S-IgAs

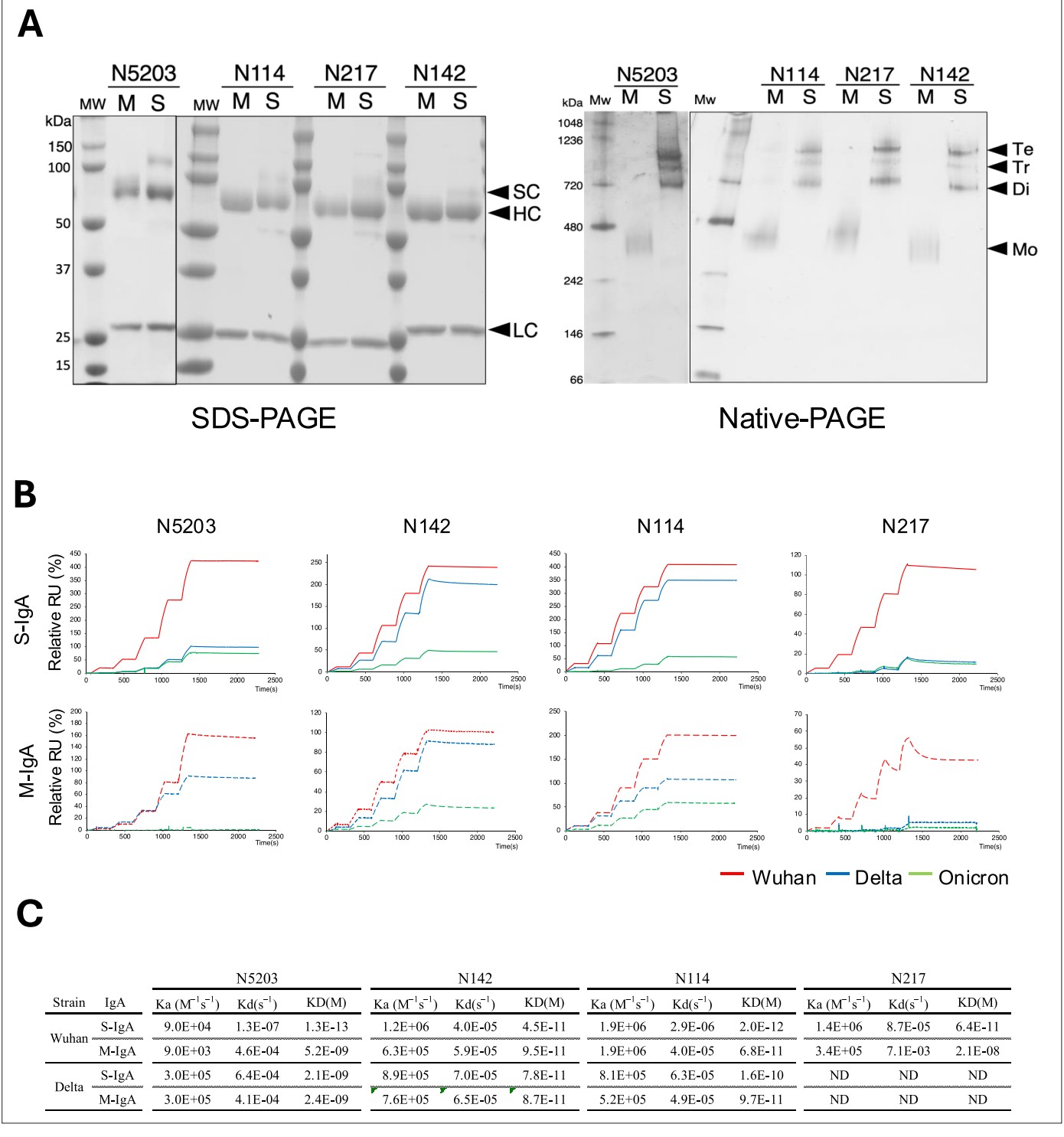

**Figure 4.** Comparison of reactivity between monomeric and multimeric immunoglobulin As (IgAs). (**A**) Production of recombinant monomeric IgAs (M-IgAs) and secretory IgAs (S-IgAs). Recombinant M-IgA and S-IgA purified from the culture supernatant of CHO cells were subjected to SDS-PAGE and Blue native-PAGE analysis. Bands corresponding to a monomer (M), dimer (Di), trimer (Ti), and tetramer (Te) are shown. H, α heavy chain; L, light chain; J, J-chain; SC, secretory component; M, M-IgA; S, S-IgA. (**B**) Binding dynamics of M-IgAs and S-IgAs to Wuhan, Delta, or Omicron spike protein by surface plasmon resonance (SPR). The S-IgAs used are a mixture of dimers, trimers, and tetramers. The observed values reflect the average affinity of the S-IgAs. The curves shown are representative of two or three determinations. RU, resonance units. Representative data from two independent experiments are shown. (**C**) The table shows the association (ka) ($M^{-1}s^{-1}$), dissociation (kd) ($s^{-1}$) rate constants and apparent equilibrium dissociation constants ($K_D$) expressed as the mean of two or three determinations (lower panel).

*Figure 4 continued on next page*

*Figure 4 continued*

The online version of this article includes the following source data for figure 4:

**Source data 1.** Raw uncropped native and SDS-PAGE data showing the Recombinant M-IgAs and S-IgAs.

**Source data 2.** Recombinant M-IgA and S-IgA raw native and SDS-PAGE data were labeled with their associated bands prior to sectioning.

showed only a small weight loss (5% with N5203, 8% with N142, and 5% with N217) (*Figure 6A*). Analysis of viral load at day 3 by qPCR for SARS-CoV-2 nucleoprotein (N2) RNA showed that some hamsters treated with N5203 S-IgA or N217 S-IgA showed a trend of decreased virus production in the nasal turbinate compared to the control. However, this did not reach statistical significance. These antibodies did not reduce the viral load in the lungs. These results indicate that the prophylactic nasal administration of S-IgA confers a protective effect against SARS-CoV-2, but does not lead to the elimination of the virus from the animals.

## Discussion

In this study, we generated hundreds of monoclonal antibodies from nasal mucosa and nonmucosal tissues, providing molecular evidence that intranasal immunization can induce both mucosal and systemic antibody responses. We also generated M-IgA and S-IgA with the same antigen recognition site for the Spike protein and compared their antiviral activities in vitro. SARS-CoV-2 research has mainly focused on anti-RBD neutralizing antibodies, while the role of non-neutralizing antibodies has not been adequately analyzed. We demonstrated that non-neutralizing M-IgAs, comprising 70% of the nasal IgA repertoire, showed strong neutralizing activity when expressed as S-IgAs. Nasal delivery of S-IgA offered potential prophylaxis against SARS-CoV-2 infection. The ability of nasal immunization to induce various neutralizing S-IgAs that recognize different epitopes may be a significant advantage not seen with injectable vaccines. Our data highlight the key role of S-IgA in the protective effect of mucosal immunity, which may be useful for better understanding how intranasal vaccines can help protect against SARS-CoV-2 infection.

It has been shown that the mucosal route of immunization elicits immune responses at local sites and systemic immune responses (*Lapuente et al., 2021*). Our analysis demonstrated that the spleen possessed a larger population of antigen-specific plasma cells that expressed potentially nasal-derived IgAs and IgGs. These results suggest that the B cells activated in the nasal immune system give rise to plasma cells that reside in the nasal mucosa and produce S-IgA, while the B cells also migrate to the spleen and produce M-IgA and IgG. The role of the spleen in the induction of potentially nasal-derived antibodies by intranasal immunization is yet to be determined. The antigen-specific plasma cells in the spleen may respond in cases of systemic infection by supplying M-IgAs and IgGs into the bloodstream or protecting the body against subsequent infection (*Afkhami et al., 2022*; *Sheikh-Mohamed et al., 2022*). Although bone marrow and the intestines are reservoirs for IgA-producing B cells, we could not detect antigen-specific plasma cells in these tissues. The molecular mechanisms regulating potentially nasal-derived B-cell migration into the spleen but not into bone marrow and Peyer's patches remain to be determined. The extent of functional and reactive differences between M-IgA and IgG sharing the same V gene remains to be determined in this paper. Recent studies on replacing the Fc domain of IgG with IgA have shown conflicting data, with some antibodies being enhanced, while others were weakened. This suggests that the degree of increase in antibody reactivity and functionality is not solely correlated with the Fc domain but also depends on the selected epitopes and paratopes (*Tudor et al., 2012*; *Muramatsu et al., 2014*; *Ejemel et al., 2020*; *Saito et al., 2019*).

As illustrated in *Figure 7*, four types of antibodies exhibited different reactions between M-IgA and S-IgA regarding spike protein binding, ACE2-blocking, and virus neutralization.

N5203 M-IgA, categorized as Type 1, binds to the $RBD_{Wuhan}$ and $RBD_{Delta}$, inhibits ACE2-binding to these RBDs, and effectively neutralizes Wuhan and Delta pseudotyped viruses. The multimeric form of this antibody had a minimal effect on ACE-blocking and neutralization breadth despite the increased affinity for $Spike_{Wuhan}$. This suggests that the increase in valency does not necessarily correlate with an increase in functionality. N142 M-IgA, categorized as Type 2, binds to the $RBD_{Wuhan}$ and $RBD_{Delta}$, inhibiting ACE2-binding to these RBDs, yet fails to neutralize the viruses. Interestingly, its S-IgA exhibited neutralization activity against both pseudotyped viruses without enhancing spike binding

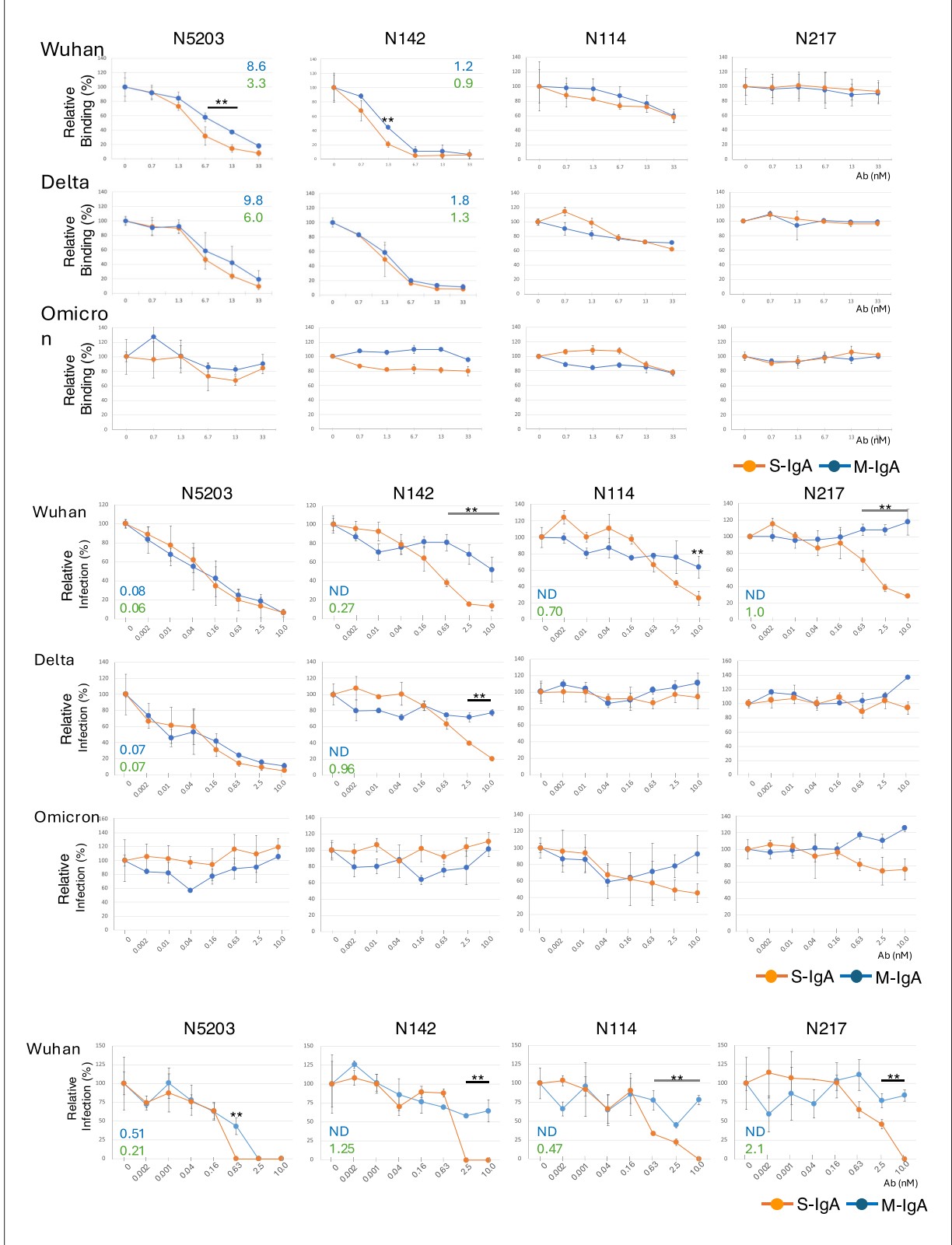

**Figure 5.** Multimerization facilitates the neutralization activity of non-neutralizing monomeric immunoglobulin As (M-IgAs). (**A**) Graphs of the competitive enzyme-linked immunosorbent assay (ELISA) results showing the binding of biotinylated angiotensin-converting enzyme-2 (ACE2) to the immobilized Wuhan, Delta, or Omicron receptor binding domain (RBD) in the presence of antibodies. The results are expressed as the mean ± SD of three technical replicates. The figure reports values from a single experiment. The IC$_{50}$ values of the indicated antibodies that inhibit the RBD-

*Figure 5 continued on next page*

*Figure 5 continued*

ACE2 interaction are shown in the diagrams. (**B**) Comparison of neutralization activity between M-IgA and secretory IgA (S-IgA) against severe acute respiratory syndrome coronavirus 2 (SARS-CoV-2) pseudotyped viruses. Neutralization curves of the indicated antibody against pseudotyped viruses bearing spike proteins of Wuhan, Delta, or Omicron are shown. Pseudotyped viruses preincubated with antibodies at the indicated concentrations were used to infect VeroE6 cells, and luciferase activities in cell lysates were measured at 20 hr post-transduction to calculate infection (%) relative to nonantibody-treated controls. The results are expressed as the mean ± SD of three technical replicates. The $NT_{50}$ values of the indicated antibodies are shown in the diagrams. Antibodies that did not reach >70% inhibition at the highest concentration tested were listed as data not determined (ND). (**C**) Comparison of neutralization potential between M-IgA and S-IgA against authentic SARS-CoV-2 BA.1. The neutralizing potential of the antibody was determined using a reverse transcription polymerase chain reaction (RT-PCR)-based SARS-CoV-2 neutralization assay. VeroE6 cells preincubated with authentic SARS-CoV-2 BA.1 virus were incubated with the indicated antibodies at various concentrations. The virus in the cell culture medium was measured at 48 hr post-transduction to calculate infection (%) relative to non-antibody-treated controls. The results are expressed as the mean ± SD of three technical replicates. The $NT_{50}$ values of the indicated antibodies are shown in the diagrams. Antibodies that did not reach >50% inhibition at the highest concentration tested are listed as ND. **p<0.01.

or ACE-blocking activity. These results suggest that the neutralizing activity of N142 S-IgA partially depends on the ACE2-blocking activity of the corresponding M-IgA, but its mechanism differs from that of N5203 S-IgA, whose monomeric form shows cross-neutralizing activity. Notably, two types of non-ACE2-blocking, non-neutralizing M-Abs (N114 and N217), expressed as S-IgA, gained the ability to neutralize the Wuhan but not the Delta pseudotyped viruses. These findings indicate that the two S-IgAs exert their neutralizing activity through mechanisms apart from the general ACE2-RBD axis.

Recent reports show that antibodies that induce inter- or intraspike cross-linking can inhibit viral binding or shedding by the host cell through steric hindrance or cause conformational changes in the spike proteins (*Galimidi et al., 2015*; *Jackson et al., 2022*; *Klein and Bjorkman, 2010*). Such inter- and intraspike cross-linking may be limited in M-IgAs and IgGs if the spike protein density on the virus surface is low or if the epitopes are unfavorably located. We hypothesize that S-IgA could exert the above effects through its multivalent arms, a mode of action that is hard to achieve with M-IgA, and factors such as valency, epitope selection, antibody binding angle, and the bulkiness of the Fc are involved in this process (*Okuya et al., 2020a*; *Callegari et al., 2022*).

Intranasal delivery of monoclonal antibodies was shown to confer viral protection in a prophylactic setting in animal models (*Haga et al., 2021*; *Zhou et al., 2023*; *Marcotte et al., 2024*). We have shown that hamsters receiving S-IgA intranasally are protected against weight loss by SARS-CoV-2 infection. However, this treatment did not decrease viral load in the nasal turbinates and lungs. Since the S-IgAs used in this experiment represent the general antibodies found in the nasal mucosa rather than the potent antibodies selected through screening, they lacked sufficient capacity to prevent viral infections completely. However, when these antibodies exist as polyclonal antibodies on the nasal mucosa, they may be more effective in preventing infections.

Our study has limitations, including that the number of antigen-specific plasma cells isolated from the nasal mucosa of immunized mice was insufficient for a comprehensive analysis of systemic and mucosal immune responses. In this study, antibodies that did not share V-(D)-J with nasal clones were found in the spleen, lungs, and blood of immunized mice. This finding was probably due to the limited number of clones that were isolated from the nose. However, we cannot exclude the possibility that these clones differentiated from local naïve B cells. While S-IgA has many possible advantages in antibody therapy, S-IgA has the property of aggregating. Therefore, it encounters challenges in expression and stabilization. Addressing these issues is crucial for large-scale production for in vivo experiments.

## Study approval

All experiments were performed in accordance with relevant guidelines and regulations. The Committee on Animal Experimentation at the University of Toyama approved the animal experimental protocols, which were conducted using project license A2017eng-1. All experiments involving live SARS-CoV-2 followed the approved standard operating procedures in the Biosafety Level 3 facility at the University of Toyama and Toyama Institute of Health.

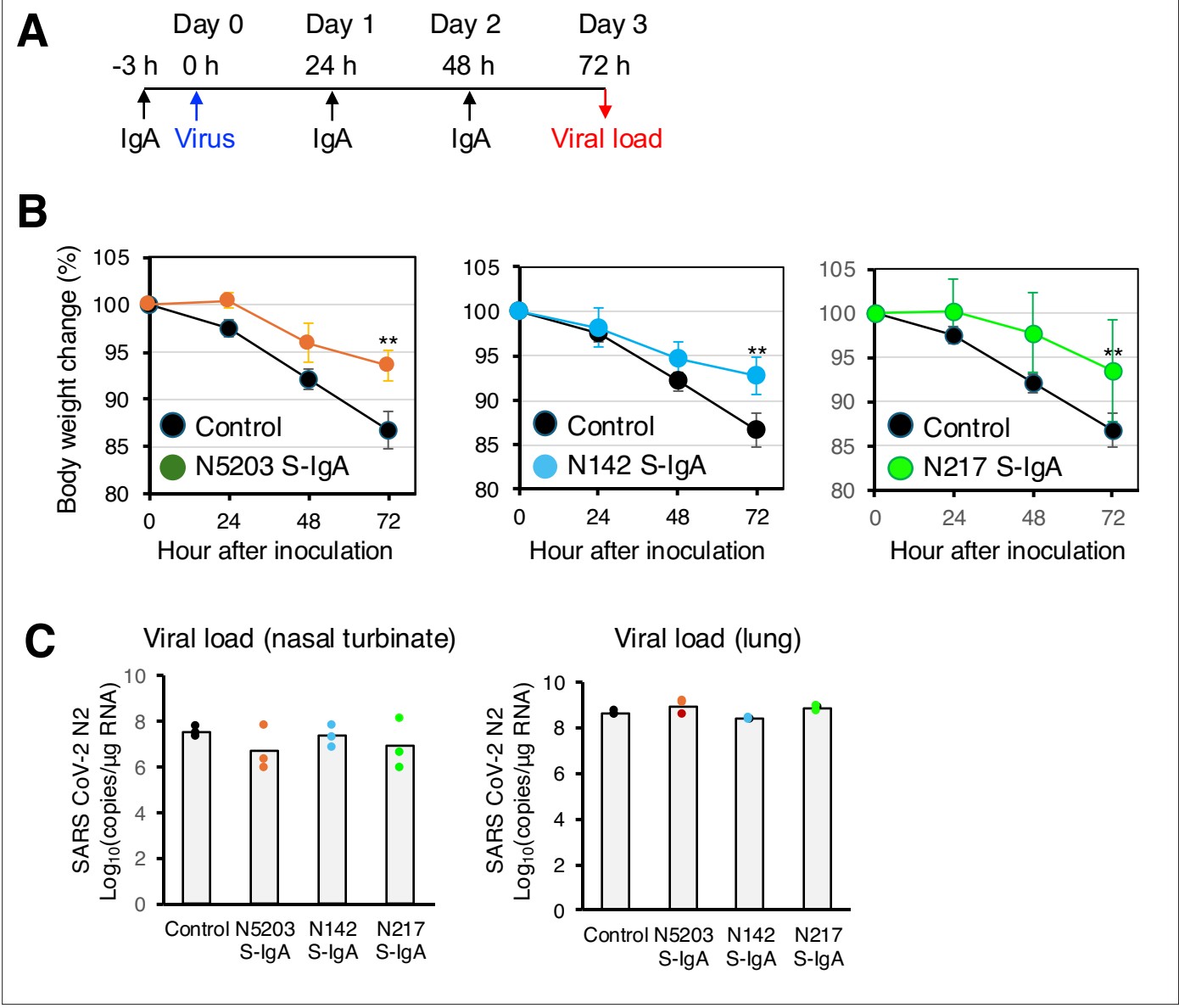

**Figure 6.** Intranasal administration of secretory immunoglobulin A (S-IgA) suppresses severe acute respiratory syndrome coronavirus 2 (SARS-CoV-2) infection in Syrian hamsters. (**A**) Experimental schedule. Three groups of hamsters received a single intranasal dose of 1.0 mg/kg of S-IgA 3 hr before infection (–3) for pre-exposure prophylaxis and at 24 hr (24) and 48 hr post-infection for early treatment, respectively. Control hamsters (n=3) received phosphate-buffered saline (PBS) at the same dose. The figure reports values from a single experiment. On day 0, each hamster was intranasally challenged with the Wuhan SARS-CoV-2 virus ($6 \times 10^5$ median tissue culture infectious dose). (**B**) Hamster body weights were recorded hourly (0, 24, 48, and 72 hr), and weight loss was defined as the percentage loss from 0 hr. Data represent the mean value ± SD at the indicated time points (n = 3) at the indicated time points and were analyzed using a Kruskal-Wallis one-way ANOVA (*p<0.05 and **p<0.01) . (**C**) Animals were euthanized 72 hr post-infection, and RNA was extracted from the nasal turbinates and lungs. The SARS-CoV-2 viral load was analyzed using quantitative reverse transcription polymerase chain reaction (qRT-PCR) targeting the SARS-CoV-2 nucleoprotein. Assays were normalized relative to total RNA levels. Data represent the mean value (n = 3) analyzed in one experiment.

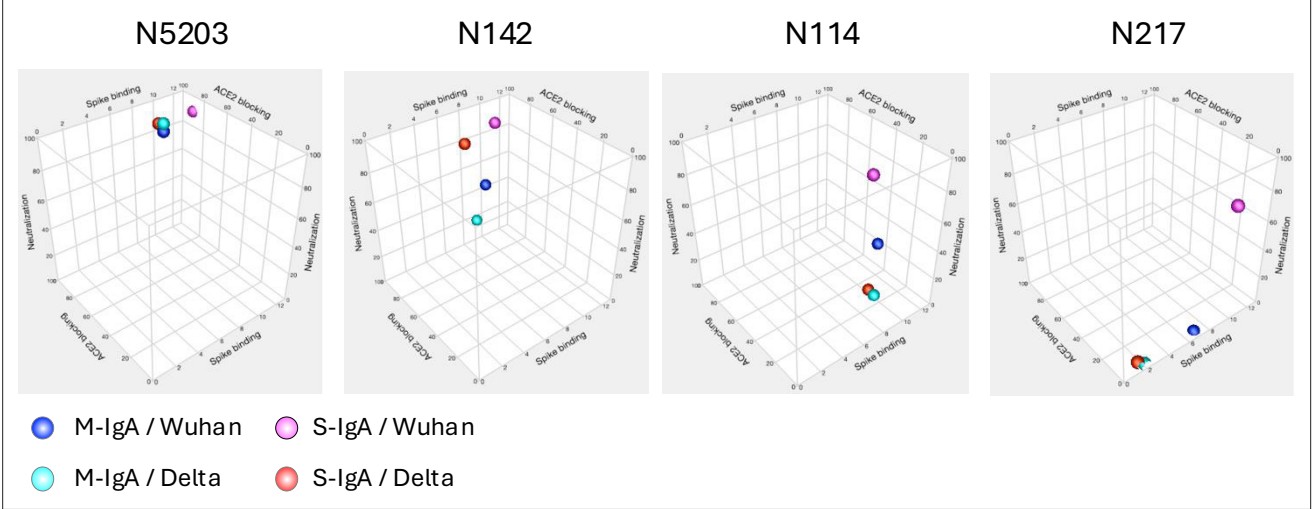

**Figure 7.** The affinity, angiotensin-converting enzyme-2 (ACE2) inhibitory activity, and the in vitro neutralizing activity of the indicated antibodies are illustrated in a 3D scatter plot.

## Materials and methods
### Materials

The materials used in this study can be found in the Key resources table. VeroE6/TMPRSS2 cells were purchased from the Japanese Collection of Research Bioresources (JCRB) Cell Bank (the National Institute of Biomedical Innovation, Health, and Nutrition, Osaka, Japan) (JCRB1819). CHO-S cells were purchased from Thermo Fisher Scientific (A29127). All cell lines tested negative for mycoplasma contamination, and the suppliers verified their authenticity through STR profiling. Female ICR mice (SLC:ICR) and female Syrian golden hamsters (SLC:Syrian) were purchased from Japan SLC, Inc (Tokyo, Japan).

**Key resources table**

| Reagent type (species) or resource | Designation | Source or reference | Identifiers | Additional information |
|---|---|---|---|---|
| Gene (*Mus musculus*) | IgHa | NCBI | AK136471 | |
| Gene (*Homo sapiens*) | IGHA2 | NCBI | AL389978 | |
| Gene (*H. sapiens*) | JCHAIN | NCBI | NM_144646 | |
| Gene (*H. sapiens*) | PIGR | NCBI | NM_002644.4 | |
| Strain, strain background (*M. musculus*) | ICR | SLC | SLC:ICR | Six-week-old female |
| Strain, strain background (*M. musculus*) | Syrian hamster | SLC | SLC:Syrian | Six-week-old female |
| Cell line (*H. sapiens*) | VeroE6/TMPRSS2 | JCRB, RRID:CVCL_YQ49 | JCRB1819 | Infection |
| Cell line (*Cricetulus griseus*) | CHO-S | Thermo Fisher, RRID:CVCL_5J31 | A29127 | Antibody expression |
| Biological sample (virus) | SARS-CoV-2 virus | NIID | 2019-nCoV/Japan/TY/WK-521/2020 | Wuhan |
| Antibody | DyLight488-labeled goat anti-mouse IgA alpha chain | Abcam | ab97011 | 1:250 |
| Antibody | Rat anti-mouse CD138 | BioLegend, RRID:AB_10915989 | 142503 | 1:250 |

*Continued on next page*

*Continued*

| Reagent type (species) or resource | Designation | Source or reference | Identifiers | Additional information |
|---|---|---|---|---|
| Antibody | DyLight488-labeled goat anti-mouse IgG H&L | Abcam | ab96871 | 1:250 |
| Recombinant protein | Biotinylated SARS-CoV-2 S1 protein NTD, His, Avitag | Acro Biosystems | S1D-C52E2 | Wuhan |
| Recombinant protein | SARS-CoV-2 (COVID-19) S protein RBD, His Tag | Acro Biosystems | SPD-C52H3 | Wuhan |
| Recombinant protein | SARS-CoV-2 S protein RBD (K417N, E484K, N501Y), His Tag | Acro Biosystems | SPD-C52Hp | B.1.351/Beta Variant |
| Recombinant protein | SARS-CoV-2 Spike RBD (L452R, E484Q), His Tag | Acro Biosystems | SPD-C52Hv | B.1.617.1 (Kappa) |
| Recombinant protein | SARS-CoV-2 (COVID-19) S protein RBD (L452R, T478K), His Tag | Acro Biosystems | SPD-C52Hh | B.1.617.2/Delta |
| Recombinant protein | Biotinylated Human ACE2 | Sino Biological | 10108-H08H-B | |
| Recombinant protein | SARS-CoV-2 S protein, His Tag, Super stable trimer | Acro Biosystems | SPN-C52H9 | Wuhan |
| Recombinant protein | Biotinylated SARS-CoV-2 S protein (D614G), His, Avitag, Super stable trimer | Acro Biosystems | SPN-C82E3 | Wuhan |
| Recombinant protein | Biotinylated SARS-CoV-2 Spike Trimer, His, Avitag (B.1.1.529/Omicron) | Acro Biosystems | SPNC82EE | B.1.1.529/Omicron |
| Recombinant protein | Biotinylated SARS-CoV-2 (COVID-19) S1 protein (D614G), His, Avitag | Acro Biosystems | S1N-C82E3 | Wuhan |
| Recombinant protein | Streptavidin Protein, DyLight 650 | Thermo Fisher | 84547 | Plasma cell isolation |
| Recombinant DNA reagent | pET-mIgA | This paper | | Antibody expression plasmid |
| Recombinant DNA reagent | pET-mIgK | This paper | | Antibody expression plasmid |
| Commercial assay or kit | THUNDERBIRD Probe One-step qRT-PCR Kit | TOYOBO | QRZ-101 | qPCR |
| Commercial assay or kit | SureBlue/TMB peroxidase substrate | Sera care | 5120-0059 | ELISA |
| Commercial assay or kit | N2 primer/probe set | Nihon Gene Research Laboratories | 283-34201 | qPCR |
| Commercial assay or kit | CHOgro High Yield Expression System | TakaraBaio/Mirus Bio | MIR 6260 | Antibody expression |
| Commercial assay or kit | Dynabeads mRNA Purification kit | Veritas | DB61006 | |
| Commercial assay or kit | NativePAGE Bis-Tris Gel System | Thermo Fisher | BN1001BOX | |
| Chemical compound, drug | ER-Tracker BlueWhite DPX | Thermo Fisher | E12353 | Plasma cell isolation |
| Chemical compound, drug | Cholera toxin | Fujifilm | 030-20621 | Adjuvant |
| Software, algorithm | GENETYX | NIHON SERVER | | Antibody sequence analysis |
| Software, algorithm | Sequencher | Gene Codes | Version 5.4.6 | Antibody sequence analysis |
| Software, algorithm | RStudio | posit | Version 2022.12.0+353 | Antibody lineage analysis |

*Continued on next page*

*Continued*

| Reagent type (species) or resource | Designation | Source or reference | Identifiers | Additional information |
|---|---|---|---|---|
| Software, algorithm | Alakazam | https://www.rdocumentation.org/packages/alakazam/ | Versions/1.2.1 | Antibody lineage analysis |
| Other | Peptide M Agarose | Thermo Fisher | gel-pdm-2 | IgA purification |
| Other | TiterMax adjuvants | Merck Sigma-Aldrich | 12352203 | Immunization |
| Antibody | Goat Anti-Mouse IgA alpha chain (HRP) | abcam | ab97235 | ELISA |

## Immunization

Anesthetized 6-week-old female ICR mice were intranasally immunized with 10 µg of Spike$_{Wuhan}$ or PBS with 1 µg of cholera toxin in 10 µl of PBS by using a pipette to deliver the fluid dropwise into each nostril a total of three times at 3-week intervals. One week after the final immunization, *mice* were sacrificed by $CO_2$ inhalation, and blood samples were collected from the inferior vena cava. To avoid contamination with circulating blood lymphocytes, the mice were perfused with 2 ml of PBS via the heart. Then, the lungs, the spleen, bone marrow, and Peyer's patches were dissected, and single-cell suspensions were obtained by mincing the tissue using a 100 µm nylon mesh. The mice were decapitated at the larynx. After the removal of the facial skin from the head, the nose was separated from the rest of the head. A pipette tip was inserted through the pharyngeal opening into the choana, and then two consecutive volumes of 250 µl of PBS were gently perfused, and the nostril fluid was collected in a tube. The nasal lavage fluids were centrifuged, and supernatants were stored at −80°C until assayed. The isolated nose was cut into two along the midline; the layer of epithelium was mechanically removed from the nasal septum by gently rubbing the sample with a needle under a stereoscopic microscope, and the tissues were then mechanically shredded. Single-cell suspensions containing lymphocytes were isolated by using Lympholyte-M (Cedarlane, Ontario, Canada).

## ELISA

ELISA to evaluate antibody binding to the Spike Trimer, RBDs, or NTD was performed by coating Nunc MaxiSorp flat-bottom high-binding 96-half-well plates (Thermo Fisher Scientific, MS, USA) with 50 µl per well of a 0.1 µg/ml protein solution in PBS overnight at 4°C. Plates were washed three times with PBS and incubated with 170 µl per well Blocking One solution (Nakarai, Tokyo, Japan) for 1 hr at room temperature. Serially diluted antibody, serum, or nasal lavage fluid was added to PBST (1× PBS with 0.1% Tween-20) and incubated for 1 hr at room temperature. The plates were washed three times with PBST and then incubated with anti-mouse IgG or IgA secondary antibody conjugated to horseradish peroxidase (HRP) (Abcam, Cambridge, UK) in PBST for 1 hr at room temperature. After washing with PBST three times, the plates were developed by the addition of the SureBlue/TMB peroxidase substrate and stop solution (KPL, MS, USA). Absorbance was measured at 450 nm with an ELISA microplate reader. Protein concentration in serum and nasal lavage fluid is determined by the BCA Protein Assay Kit (TaKaRa, Tokyo, Japan).

## ACE2 blocking assay

Nunc MaxiSorp plates were coated with SARS-CoV-2 RBD at 50 ng per well and incubated overnight at 4°C. After blocking with Blocking One solution for 1 hr at room temperature, the serially diluted antibody mixed with 5 ng of biotinylated ACE2 in 100 µl of PBST was transferred to the plate in triplicate. After incubation for 1 hr at room temperature, the assay plate was washed with PBST three times, and 100 µl of streptavidin-HRP (Abcam, Cambridge, UK) diluted 1:5000 in PBST was transferred to each well and incubated for 30 min. After three washes, the plate was developed with streptavidin-HRP and SureBlue/TMB peroxidase substrate.

## Plasmid construction

The DNA fragments encoding the human IgA2 constant region, human J-chain, and extracellular domain of human pIgR were amplified by PCR. The IgA2 constant region was replaced with the IgG1

constant region of pJON-mIgG or pETmIgA to make pJON-hIgA and pET-hIgA, respectively (*Kurosawa et al., 2012*). The human J-chain was replaced with the DsRed2 gene of pDsRedN1 (Takara Bio, Shiga, Japan). The pIgR gene was inserted into the pEF-Myc-His vector (Thermo Fisher Scientific, MS, USA).

## Isolation of antigen-specific plasma cells

Isolation of antigen-specific plasma cells was performed as described previously with slight modifications (*Kurosawa et al., 2012*). Cells were stained with PE-labeled anti-mouse CD138, DyLight-650-labeled S1, and DyLight488-labeled anti-mouse IgA or anti-mouse IgG at 4°C for 30 min with gentle agitation. After washing with PBS, the cells were suspended in PBS containing ER-Tracker Blue-White DPX (ER-Tracker) and subsequently analyzed by FACS. The forward-versus-side-scatter lymphocyte gate (R1) was applied to exclude dead cells. The plasma cells (IgA+ or IgG+, CD138+, ER-Tracker[high], R2 gate) were further subdivided into fractions according to their binding of DyLight-650-labeled S1 domain of SARS-CoV-2 Spike protein to define antigen-specific plasma cells (IgA+, ER-Tracker[high], S1+). Single-cell sorting was performed using a JSAN Cell Sorter that was equipped with an automatic cell deposition unit (JSAN, Kobe, Japan) with DyLight488-labeled antibodies against IgA or IgA monitored in the FL-l channel, PE-labeled CD138 in the FL-2 channel, ER-Tracker in the FL-7 channel, and DyLight650-labeled S1 in the FL-6 channel.

## Monoclonal antibody generation

Molecular cloning of VH and VL genes from single cells was performed by 5'-RACE PCR as previously described. For the first antibody screening, the PCR-amplified VHa and VL genes were joined to pJON-hIgA and pJON-mIgK to make full-length immunoglobulin alpha heavy and kappa light chain genes by TS-jPCR, respectively (*Yoshioka et al., 2011*). M-IgA was expressed by transfecting a pair of alpha heavy and kappa light chain genes into FreeStyle CHO-S cells that were grown in a 24-well plate according to the manufacturer's protocol (CHOgro High Yield Expression System, Takara Bio, Shiga, Japan). For large-scale antibody production, the PCR-amplified alpha heavy and kappa light chain genes were inserted into pET-hIgA and pET-mIgK by TS-HR, respectively (*Kurosawa et al., 2011*).

M-IgA was expressed by cotransfecting plasmids encoding the IgA gene (5 µg) and IgK gene (5 µg) into FreeStyle CHO-S cells ($2.0×10^5$ cells/10 ml). S-IgA was expressed by cotransfecting plasmids encoding the IgA gene (5 µg), IgK gene (5 µg), J-chain (1 µg), and pIgR gene (1 µg) into FreeStyle CHO-S cells ($2.0×10^5$ cells/10 ml). M-IgA was purified by using Peptide M Agarose (Thermo Fisher Scientific). S-IgA was purified by two-step chromatography using the Capturem His-Tagged Purification Kit (Takara Bio, Shiga, Japan) followed by size exclusion chromatography (Cytiva, AKTA go, MS, USA). The purified antibodies were analyzed by SDS-PAGE and native polyacrylamide gel electrophoresis on NuPAGE 4–12% Bis-Tris gels (Thermo Fisher Scientific MS, USA).

## Antibody binding kinetics by SPR

The binding kinetics and affinity of monoclonal antibodies to the SARS-CoV-2 spike trimer were analyzed by SPR (Biacore T100, GE Healthcare). Specifically, a biotinylated spike trimer was covalently immobilized to an SA Sensor Chip for a final RU of approximately 200 and interacted with S-IgA containing a mixture of dimers, trimers, and tetramers or M-IgA at various concentrations (0.3, 1.0, 3.0, 9.0, and 27 nM of each antibody). SPR assays were run at a 30 µl/min flow rate in HEPES buffer. The dissociation phase was monitored for 5 min. The sensograms were fitted into a two-component model with BIAevaluation software (GE Healthcare).

## Neutralization activity of monoclonal antibodies against the pseudotyped SARS-CoV-2 virus

VeroE6/TMPRSS2 cells were incubated with serially diluted antibodies and pseudotyped virus possessing the spike protein of the Wuhan, Delta, Omicron strains or vesicular stomatitis virus and cultured for 48 hr at 37°C. After exposure to the virus-antibody mixture, the infectivity of the pseudotyped viruses was determined by measuring the luciferase activities using a PicaGene Luminescence Kit (Fujifilm Wako, Osaka, Japan) with a GloMax Navigator Microplate Luminometer (Promega, WI, USA).

### Neutralization assay using an authentic virus strain

The neutralizing activity of monoclonal antibodies against an authentic Wuhan SARS-CoV-2 strain was determined by a neutralization test in a biosafety level 3 laboratory at the Toyama Institute of Health as previously described (*Ozawa et al., 2022*). VeroE6/TMPRSS2 cells plated at $2 \times 10^4$ cells in each well of 96-well plates were infected with the Wuhan SARS-CoV-2 strain at a multiplicity of infection of 0.001 per cell in the presence of serially twofold diluted monoclonal antibodies for 1 hr. After discarding the culture supernatants, cells were cultured for 24 hr with DMEM containing 10% FBS in the presence of the indicated concentration of monoclonal antibodies. The viral infectious dose was determined by the level of viral genomic RNA in the culture supernatant, which was measured using a real-time PCR assay with a SARS-CoV-2 direct detection RT-qPCR kit (Takara Bio, Siga, Japan). The $IC_{50}$ was calculated by IC50 Calculator (https://www.aatbio.com/tools/ic50-calculator) and represented the neutralization titer.

### Hamster models

Six-week-old female Syrian hamsters were purchased from Japan SLC Inc (Shizuoka, Japan). The hamsters were maintained under a 12 hr light-dark cycle and had unrestricted access to food and water in the Division of Animal Resources and Development at the University of Toyama. All animal procedures were approved by the Animal Experiment Committee of the University of Toyama (protocol number: A2023MED-10). Hamsters were intranasally administered 1 mg/kg of IgA in 100 µl of PBS (50 µl to each nostril) under anesthesia 3 hr before intranasal inoculation of $6.0 \times 10^5$ TCID50 in 100 µl of PBS (50 µl to each nostril) of SARS-CoV-2 (2019-nCoV/Japan/TY/WK-521/2020) under anesthesia. At 24 and 48 hr after SARS-CoV-2 inoculation, 1 mg/kg of IgA was administered intranasally under anesthesia. Both SARS-CoV-2 inoculation and IgA administration were performed under anesthesia with a mixture of midazolam (2.4 mg/kg) (Maruishi Pharmaceutical Co., Ltd., Osaka, Japan), butorphanol tartrate (3 mg/kg) (Meiji, Tokyo, Japan), and medetomidine hydrochloride (0.18 mg/kg) (ZENOAQ, Fukushima, Japan), and the hamsters were awakened from anesthesia with atipamezole hydrochloride (0.18 mg/kg) (ZENOAQ) after the inoculation or administration. The hamsters were anesthetized and euthanized 72 hr after inoculation with SARS-CoV-2.

### RNA extraction and real-time RT-PCR

The nasal turbinates and lungs were collected in ISOGEN (Nippon Gene, Tokyo, Japan) and homogenized with 1.4 mm Ceramic Beads (Thermo Fisher Scientific, Hampton, NH, USA) using BeadMill 24 (Thermo Fisher Scientific) at a speed of 4 m/s for 3 min and stored at −80°C until subsequent use. Total RNA was extracted using ISOGEN according to the manufacturer's instructions. Real-time reverse transcription polymerase chain reaction (RT-PCR) was performed by amplifying the SARS-CoV-2 N2 gene using QuantStudio 6 Pro Real-Time PCR System (Thermo Fisher Scientific) and THUNDERBIRD Probe One-step qRT-PCR Kit (TOYOBO, Osaka, Japan) with the N2 primer/probe set (Nihon Gene Research Laboratories, Miyagi, Japan). The absolute copy number of the SARS-CoV-2 N2 gene was determined by serial dilution of RNA control (Nihon Gene Research Laboratories).

### Gene family and phylogenetic analysis of monoclonal antibodies

The antibody sequences were annotated against the IMGT mouse heavy and light chain gene database using NCBI IgBlast to determine IGHV, IGHD, IGLV, IGHJ, IGLV, and IGLJ gene annotations. Antibody clones were assigned to clonal groups using Sequencher software. The heavy and light chain variable gene arrangement and phylogenetic analyses were performed using MAFFT, a multiple alignment program, in the GENETYX sequence analysis package (https://www.genetyx.co.jp). For SHM, IGHV and IGLV sequences were aligned against representative clones of each group. Antibody clones consisting of pairs of heavy and light chain variable genes (signal sequence to FW4) were used to generate an antibody phylogenetic tree. Full-length germline sequences were reconstructed, with nucleotide additions/deletions in the junction between V-(D)-J adjusted to match the sequence of each antibody group. Within these groups, if the combined sequence of the heavy and light chains differed by more than five bases from each other, they were defined as separate clones. The sequence alignment tool Clustal Omega (https://www.ebi.ac.uk/Tools/msa/clustalo/) was used to identify deletions and insertions and to align the length of the sequence. The nucleotide lengths were aligned by adding '-' to the missing bases. The data file required by Alakazam was created in RStudio (version 2022.12.0+353), and

a phylogenetic tree was created by Alakazam (https://www.rdocumentation.org/packages/alakazam/versions/1.2.1 and https://alakazam.readthedocs.io/en/stable/). The R script was as follows: https://www.rdocumentation.org/packages/alakazam/versions/1.2.1/topics/buildPhylipLineage.

## Statistical analysis

All statistical analyses were performed using JMP statistical software (JMP Statistical Discovery, NC, USA). Unpaired Student's t-tests were used to analyze each dataset. The threshold for statistical significance was set at $p < 0.01$ (**).

## Acknowledgements

We thank past and current members of our laboratory for fruitful discussions. We also thank M Nozaki and K Takai for technical support. This research was supported by grants from the Japan Agency for Medical Research and Development (AMED) (19187977, 20333128, and 22723616) and The Toyama Pharmaceutical Valley Development Consortium (no grant number assigned) to Masaharu Isobe and Japan Society for the Promotion of Science, Japan (Grant in Aid for Scientific Research B) (22H02875) to Nobuyuki Kurosawa. The funders had no role in the study design, data collection, and analysis, publishing decisions, or the preparation of the manuscript.

## Additional information

### Funding

| Funder | Grant reference number | Author |
| --- | --- | --- |
| Japan Agency for Medical Research and Development | 19187977 | Masaharu Isobe |
| Japan Society for the Promotion of Science | 22H02875 | Nobuyuki Kurosawa |
| Japan Agency for Medical Research and Development | 20333128 | Masaharu Isobe |
| Japan Agency for Medical Research and Development | 22723616 | Masaharu Isobe |

The funders had no role in study design, data collection and interpretation, or the decision to submit the work for publication.

### Author contributions

Kentarou Waki, Hideki Tani, Eigo Kawahara, Yoshitomo Morinaga, Formal analysis, Investigation; Yumiko Saga, Takahisa Shimada, Emiko Yamazaki, Formal analysis; Seiichi Koike, Investigation; Masaharu Isobe, Funding acquisition, Project administration; Nobuyuki Kurosawa, Data curation, Formal analysis, Supervision, Funding acquisition, Validation, Investigation, Methodology, Writing – original draft, Project administration, Writing – review and editing

### Author ORCIDs

Hideki Tani ![ORCID] https://orcid.org/0000-0002-6309-277X
Seiichi Koike ![ORCID] https://orcid.org/0000-0001-8203-4498
Nobuyuki Kurosawa ![ORCID] https://orcid.org/0000-0002-1548-4541

### Ethics

All experiments were performed in accordance with relevant guidelines and regulations. The Committee on Animal Experimentation at the University of Toyama approved the animal experimental protocols, which were conducted using project license A2017eng-1.All experiments involving live SARS-CoV-2 followed the approved standard operating procedures in the Biosafety Level 3 facility at the University of Toyama and Toyama Institute of Health.

Reviewer #2 (Public review): https://doi.org/10.7554/eLife.88387.3.sa1
Author response https://doi.org/10.7554/eLife.88387.3.sa2

## Additional files

### Supplementary files

Supplementary file 1. Nucleotide sequences of antigen-specific antibodies and their V-(D)-J usage. Antibody sequences are organized by clone name, class, type, and group. The nucleotide sequences of the variable domains of the heavy and light chains are analyzed using IMGT/HighV-QUEST, and the best-matched IMGT reference germline alleles are provided. The somatic hypermutation (SHM) status is evaluated based on the number of mutations present in the sequences of the heavy and light chain variable domains.

MDAR checklist

### Data availability

All data generated or analysed during this study are included in the manuscript and supporting files. The nucleotide sequence data are available in *Supplementary file 1* and have been submitted to the DDBJ/EMBL/GenBank databases (accession numbers LC761310–LC761361, LC761441–LC761570, LC761965–LC762059, LC764607–LC764701, LC761389–LC761570).

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
