## [Editor Report · eLife Assessment]

This work provides **important** insights into mucosal antibody responses against SARS-CoV-2 following intranasal immunization by characterizing a large number of monoclonal antibodies at both mucosal and non-mucosal sites. The evidence supporting the claims is **solid**. The demonstrated in vitro antiviral activity of antibodies characterized provides a rationale for developing mucosal vaccines, especially if confirmed in vivo and benchmarked against antibodies generated following intramuscular vaccination.

---

## [Referee Report · Reviewer #2 (Public review)]

Summary:

Demonstrate the breadth of IgA response as determined by isolating individual antigen-specific B cells and generating mAbs in mice following intranasal immunization of mice with SARS-CoV2 Spike protein. The findings show that some IgA mAb can neutralize the virus, but many do not. Notable immunization with Wuhan S protein generates a weak response to the omicron variant.

Strengths:

Detailed analysis characterizing individual B cells with the generation of mAbs demonstrates the response's breadth and diversity of IgA responses and the ability to generate systemic immune responses.

Comments on Revision:

I have re-reviewed the paper and responses to my and other reviewers' comments. I feel the authors have adequately addressed my and other reviewer's comments.

---

## [Author Response]

The following is the authors’ response to the original reviews

**Reviewer #1 (Public Review):**
Despite evidence suggesting the benefits of neutralizing mucosa-derived IgA in the upper airway in protection against the SARS-CoV-2 virus, all currently approved vaccines are administered intramuscularly, which mainly induces systemic IgG. Waki et al. aimed to characterize the benefits of intranasal vaccination at the molecular level by isolating B cell clones from nasal tissue. The authors found that Spike-specific plasma cells isolated from the spleen of vaccinated mice showed significant clonal overlap with Spikespecific plasma cells isolated from nasal tissue. Interestingly, they could not detect any spike-specific plasma cells in the bone marrow or Peyer's patches, indicating that these nose-derived cells did not necessarily home to and reside in these locations, although the Peyer's patch is not a typical plasma cell niche - rather the lamina propria of the gut would have been a better place to look. Furthermore, they found that multimerization improves the antibody/antigen binding when the antibody is of low or intermediate affinity, but that high-affinity monomeric antibodies do not benefit from multimerization. Lastly, the authors used a competitive ELISA assay to show that multimerization could improve the neutralizing capacity of theseantibodies.The strength of this paper is the cloning of multiple IgA from the nasal mucosae (n=99) and the periphery (n=114) post-SARS-CoV-2 i.n. vaccination to examine the clonal relationship of this IgA with other sites, including the spleen. This analysis provides novel insights into the nature of the mucosal antibody response at the site where the host would encounter the virus, and whether this IgA response disseminates to othertissues.There were also some weaknesses:(1) The finding that multimerization improves binding and neutralization is not surprising as this was observed before by Wang and Nussenzweig for anti-SARS-CoV-2 IgA (authors should cite Enhanced SARS-CoV-2 neutralization by dimeric IgA. Wang et al., Sci. Transl. Med 2021, 13:3abf1555).

We have cited the paper, and the relevant sentence has been modified as follows (line 51-53); Recent studies have demonstrated that multimeric IgA is more effective and provides greater cross-protection than IgG and M-IgA (Okuya et al., 2020b) (Asahi et al., 2002) (Dhakal et al., 2018) (Asahi-Ozaki et al., 2004) (Wang et al., 2021).

In addition, as far as I can tell we cannot ascertain the purity of fractions from the size exclusion chromatography thus I wasn't sure whether the input material used in Fig. 4 was a mixed population of dimer/trimer/tetramer?

The S-IgAs used in the SPR analysis in Fig. 4 consist of a mixture of dimers, trimers, and tetramers. The observed values indicate the average affinity of the S-IgAs. Please refer to the revised version (line 278280).

(2) The flow cytometric assessment of the IgA+ clones from the nasal mucosae was difficult to interpret (Fig. 1B). It was hard for me to tell what they were gating on and subsequently analyzing without an IgA-negative population for reference.

We have updated FACS plots to illustrate the presence of IgA+ plasma cells in Fig. 1B, and the detailed gating strategy is outlined in Fig. 1B legend. Please find the relevant statements (line 115-120).

(3) While the i.n. study itself is large and challenging, it would have been interesting to compare an i.m. route and examine the breadth of SARS-CoV-2 variant S1 binding for IgGs as in Fig. 2A. Are the IgA responses derived from the mucosae of greater breadth than systemic IgG responses? Alternatively, and easier, authors could do some comparisons with well-characterized IgG mAb for affinity and cross-reactivity as a benchmark to compare with the IgAs they looked at. Overall the authors did a good job of looking at a large range of systemic vs mucosal S1-specific antibodies in the context of an intra-nasal vaccination and this provides additional evidence for the utility of mucosal vaccination approaches for reducing person-to-person transmission.

I appreciate your consideration. Recent reports indicate that some M-IgA monomers possess neutralizing activity that is equivalent to or less than that of IgGs. However, the opposite phenomenon has also been observed. These results suggest that the Fc does not merely correlate with the degree of increase in antibody reactivity or functionality. We believe the discrepancies in previous studies are due to variations in the binding modes between the epitope and paratope of each antibody clone. Nevertheless, oligomerization enhances the functionality of most monomeric antibody clones, suggesting that the multivalent S-IgA enables a mode of action that is challenging to achieve with a monomeric antibody. Please refer to the revised version (line 399-403).

Alternatively, and easier, authors could do some comparisons with well-characterized IgG mAb for affinity and cross-reactivity as a benchmark to compare with the IgAs they looked at. Overall the authors did a good job of looking at a large range of systemic vs mucosal S1-specific antibodies in the context of an intra-nasal vaccination and this provides additional evidence for the utility of mucosal vaccination approaches for reducing person-to-person transmission.

We have summarized the characteristics of the four types of nasal IgAs in Fig.7 and in the Discussion. Please refer to the revised version (line 405-422).

**Reviewer #2 (Public Review):**
Summary:This research demonstrates the breadth of IgA response as determined by isolating individual antigenspecific B cells and generating mAbs in mice following intranasal immunization of mice with SARS-CoV2 Spike protein. The findings show that some IgA mAb can neutralize the virus, but many do not. Notable immunization with Wuhan S protein generates a weak response to the omicron variant.Strengths:Detailed analysis characterizing individual B cells with the generation of mAbs demonstrates the response's breadth and diversity of IgA responses and the ability to generate systemic immune responses.Weaknesses:The data presentation needs clarity, and results show mAb ability to inhibit SARS-CoV2 in vitro. How IgA functions in vivo is uncertain.

We conducted an additional experiment using a hamster model and confirmed that S-IgAs can protect against SARS-CoV-2 infection. Please refer to the revised version (line 349-373 and 431-438).

**Reviewer #1 (Recommendations For The Authors):**
(1) Figure 1A shows antibody titers in nasal lavage fluid and serum of mice post intranasal vaccination with SARS-CoV-2 Spike protein. The Y-axis of this figure is labeled as "U/mg" however these units are not clearly defined.

The antibody titers are expressed as optical density (OD450) value per total protein in nasal lavage fluids or serum. Please find the relevant statements (line 113-114).

Furthermore, what do antibody titers in the nasal lavage fluid and serum look like post-intramuscular vaccination with the same vaccine and dose? Comparison of titers to the intramuscular route as well as to the PBS control would make this data more impactful.

We appreciate your consideration. We have not conducted experiments comparing the effects of intramuscular and intranasal administration using the same dosage and adjuvant. Cholera toxin has primarily been used as an adjuvant for nasal immunization, but it is seldom applied for intramuscular injection. We are interested in its impact on the immune compartment when using cholera toxin as an adjuvant for intramuscular injection. We plan to conduct further experiments in the future.

Lastly, in Figure 1B, the detection of nasal IgG is not shown even though the authors assess nasally-derived IgG in the spleen further into the study.

Since the number of lymphocytes that can be collected from the nasal mucosa is limited, there is an insufficient capacity to isolate IgG+ plasma cells after collecting IgA+ plasma cells. Therefore, conducting such an experiment on mice is technically challenging. A larger animal, such as rats, will be necessary to perform this experiment. Further investigation is needed to determine whether antigen-specific IgG+ plasma cells, sharing V-(D)-J with nasal IgA, can be detected in the nasal mucosa.

(2) There appears to be something amiss with the IgA stain. It is smushed up against the X-axis. Better flow cytometry profiles should be shown. Likewise in Supplemental Fig. 1A, their IgA stain appears to not be working. This must be addressed using positive and negative controls.

We have updated FACS-polts to show the IgA+ plasma cell in Fig.1B, and the detailed gating strategy is outlined in the Fig.1B legend. Please find the relevant statements on line 115-120.

(3) We do not know the purity of the samples that were subjected to SPR and since the legend of Fig. 4 is partially incorrect, it was difficult to know how this experiment was done.

The S-IgA used in the SPR analysis shown in Figure 4 is a mixture of dimers, trimers, and tetramers, and the observed values are believed to reflect the affinity of the S-IgA in the nasal mucosa. Please refer to the revised version (line 278-280).

(4) Fig. 5 results need to compare with some of the well-characterized mAb (IgG) to understand the biological significance of these neutralizing titres.

We have summarized the characteristics of the four types of nasal IgA in Fig.7 and in the Discussion. Please refer to the revised version (page 405-422).

Communication of results:(1) Authors could improve the communication of their results by introducing the vaccination protocol in the results section accompanied by a diagram of the vaccination strategy (nature of the Ag, route, and frequency). This could be Fig. 1A .

A schematic diagram of the vaccination protocol is presented in Fig.1.

(2) Care should be taken with some of the terminology. Intranasal is the accepted term but authors sometimes use "internasal". The term "immunosuppression" on page 2 could be misleading as it means something different to other audiences. The distinction when speaking about "protection from harmful pathogens" should be made between protection against infection (ie sterilizing immunity) vs protection against disease (ie morbidity and mortality). Instead of "nose", one should say "nasal". Nose-related could be rephrased as "potentially nasal-derived". P.5, line 2 didn't make sense: "IgG+ plasma cells that express nose-related IgA"...In many places, Spike is missing it's "e".

We have made the correction accordingly.

(3) Page 3: The lumping of the human and animal SARS-CoV-2 intranasal studies together is a bit misleading. Very little has worked for intranasal vaccination against SARS-CoV-2 in humans at this point in time (although hopefully that will change soon!). Authors should specify which studies were done in animals and which were done in humans.

The manuscript has been revised to include two citations on line 73-75 (Ewer et al., 2021 and Zhu et al., 2023).

(4) What is ER-tracker? It comes out of nowhere and should be explained why it was used to the reader (as well as why they used the other markers) to sort for Spike-specific PC.

ER-Tracker is a fluorescent dye that is highly selective for the endoplasmic reticulum of living cells. Because plasma cells have an expanded endoplasmic reticulum for properly folding and secreting large quantities of antibodies, using ER-Tracker along with anti-CD138 facilitates the isolation of plasma cells from lymphocytes without the need for additional antibodies. Please refer to the revised version for details. (ine 130-134).